# Sport Anxiety, Fear of Negative Evaluation, Stress and Coping as Predictors of Athlete’s Sensitivity to the Behavior of Supporters

**DOI:** 10.3390/ijerph20126084

**Published:** 2023-06-08

**Authors:** Katarzyna Gabrys, Antoni Wontorczyk

**Affiliations:** 1Doctoral School in Social Sciences, Jagiellonian University, 30-348 Krakow, Poland; 2Institute of Applied Psychology, Faculty of Management and Social Communication, Jagiellonian University, 30-348 Krakow, Poland

**Keywords:** supporters, anxiety, negative evaluation, stress, coping, sensitivity, professional athletes

## Abstract

The aim of the study was to find predictors of athlete sensitivity to the positive or negative behaviors of supporters and determine a relationship between athletes’ sensitivity to the positive and negative behaviors of supporters and traits such as anxiety and stress or the strategies used to cope with stress. The sample consisted of 171 professional athletes. The study determined three variables that are predictors of athlete sensitivity to the positive behavior of supporters (SPS), i.e., strategies of coping with stress, such as high levels of coachability, confidence and achievement motivation and low levels of freedom from worry (change R^2^ = 0.15, F of change = 9.78, *p* of change < 0.001). The predictors of sensitivity to the negative behavior of supporters (SNS) are as follows: a low level of freedom from worry and a high level of fear of negative evaluation (change R^2^ = 0.31, F of change = 38.56, *p* of change < 0.001). In the context of the athlete’s position, the predictors of SPS for midfielders are as follows: high level of susceptibility to stress and two strategies of coping with stress, i.e., peaking under pressure and concentration. For forwarders they are as follows: high level of worry and coping with stress via goal-setting, while for defenders, the ways of coping with stress are confidence and achievement motivation. In terms of SNS, for defenders, the predictors are low levels of freedom from worry, coachability, and a high level of fear of negative evaluation. Forwarders, who are sensitive to the negative behavior of supporters, are characterized by a fear of negative evaluation.

## 1. Introduction

All types of sports competitions are related to the relationship that arises between competitors and spectators during the competition. Spectators are people who may behave in various, and not always positive, ways, i.e., cheering or booing. However, athletes are not indifferent to the presence and behavior of spectators [1]. The interaction between spectators and athletes can affect the results both positively as well as negatively [2,3,4,5,6,7,8].

One of the topics of sports psychology research is the interpretation of the positive and negative emotions experienced by athletes during sports activities [9,10,11]. Positive emotions experienced by athletes during sports competitions are satisfaction [12,13], enjoyment [14,15] and eudaimonia [16]. The affective states experienced by athletes are also strongly related to the phenomenon of social affiliation. Some studies have found that when athletes experienced high levels of social affiliation, they were also more committed to training and more likely to spend a larger amount of time training than when there was no social affiliation [14,15]. A similar relationship was observed during sports competitions. Athletes with high social affiliation were found to be more engaged during sports competitions [14,15], which resulted in higher performance levels, better results, and a feeling of eudaimonia [16].

During sports competitions, athletes also experience negative affective states, one of the best-studied of which in sports psychology is anxiety [9]. Sports-related anxiety, which differs from the anxiety experienced in other life situations, is particularly important [17]. The knowledge of the characteristics of the situation in which sports competitions take place allows a better understanding of an individual’s behavior than their general level of trait anxiety [18,19]. Situation-specific trait anxiety was also an area of interest for Martens [20], who introduced the concept of competitive trait anxiety [20]. It is a tendency of the athlete to perceive competitive situations as threatening. As a consequence, a sports competition causes different intensities of anxiety for each athlete. Currently, competitive anxiety is described according to the classical distinction [21], anxiety as a trait (competitive trait anxiety) [22,23] and anxiety as a state (competitive state anxiety) [24,25]. According to Martens et al. [26], competitive state anxiety is related to the athlete’s relationship with a specific competitive situation [26]. Later, Martens et al. [27] divided competitive state anxiety into two types: cognitive anxiety and somatic anxiety. The physiological changes experienced by the athlete as a consequence of anxiety during sports competitions are called somatic anxiety [27]. Cognitive anxiety, on the other hand, is defined as worry and, therefore, the appearance of negative thoughts, the formation of negative self-esteem, and negative expectations about the outcome of sports competitions [27]. A high level of cognitive anxiety will result in low performance levels for athletes, and therefore, poor results. There is a negative, linear relationship between cognitive anxiety and performance [27]. However, increased somatic anxiety will facilitate good performance during the competition, but only up to a certain point, i.e., when the performed task reaches the optimal level for the athlete [28,29]. Once the level of optimal functioning is exceeded, the quality of performance will start to deteriorate, even if the athlete’s somatic anxiety continues to increase [30]. This phenomenon is well described in sport psychology by the Individual Zones of Optimal Functioning (IZOF) model [31,32].

However, the findings of other researchers, based on the assumptions of the multidimensional theory of anxiety [27], have not fully confirmed its accuracy (see [33] for review). The most serious criticism of this theory is that it assumes that cognitive anxiety will always have a negative impact on performance. As the research [34] shows, the quality of a performance depends not so much on the level of cognitive or somatic anxiety but on how it is interpreted by the individual. This illustrates the extent to which the relationship between the perception and understanding of anxiety and sport performance is individualized. The study of an athlete’s tendency to experience somatic and cognitive anxiety reactions in sports-related situations was the focus of research conducted by Smith et al. [35]. They pointed out the necessity of making a distinction between somatic and cognitive anxiety in relation to anxiety as a trait [35]. This allows the prediction of an athlete’s tendency to experience a certain level of somatic and cognitive anxiety in a sports-related situation [35].

A factor that also causes anxiety in athletes is the presence and evaluation of spectators, the observers of the competition [36]. Leary [37] thinks that competitive anxiety can be considered to be a type of social anxiety since competitions imply presenting oneself in front of an audience, i.e., a large group of people. Competitors are aware that they are being judged by the spectators and that it depends on their performance as to whether their evaluation is positive or negative. However, when an athlete anticipates that he or she will be judged disparagingly by others (e.g., coach, friends, other competitors), this anticipation causes subjective psychological distress and means that the athlete experiences fear of a negative evaluation (FNE) [6,38,39].

In addition to anxiety, sports competitions are also associated with stress [40,41]. The ability to locate stressful situations correctly is extremely important for the athlete. The Transactional Model of Stress and Coping [42,43] can be used to detect challenging situations in an athlete’s life. This allows us to find out how the athlete perceives and interprets such situations. It is also important to know the range of coping styles that the athlete can use to deal with stress. In their research, Smith et al. [44] distinguished seven coping strategies that are used by athletes when they perform. These strategies are as follows: coping with adversity, coachability, concentration, confidence and achievement motivation, goal-setting, peaking under pressure, and freedom from worry. Athletes usually use multiple coping strategies to deal with stress and very often mix them together [45]. It is worth noting that research so far has not shown which strategy is optimal and can be used to help achieve a better performance. Some researchers indicate that it depends on the dispositional traits of the individual [46].

The concept of athletic performance in sports psychology has also been explained according to social facilitation theory. It has been shown that the presence of spectators during the competition contributes to the better performance of athletes [47]. Competitors are professionals, so the presence of others should have a positive impact on their performance [48]. However, experimental studies relating to an athlete’s ability to complete a task that requires concentration and motor skills show that even professional athletes will perform an identical task less well in the presence of spectators than without them [49]. Furthermore, spectators are not just observers of the spectacle, and their behavior is not always related to cheering the athletes, as shown in the supporter classifications [50,51]. A sports spectacle is a complex situation in which players, supporters, and referees interact with one another [52]. An athlete’s reaction will depend on what their beliefs are about that particular situation [53]. These beliefs can be categorized into rational and irrational beliefs [54,55]. Rational beliefs (flexible, non-extreme, and logical) lead to mental health, whereas irrational beliefs (rigid, extreme, and illogical) lead to subjective psychological distress [53,56]. Studies carried out in recent years show that athletes with high levels of social anxiety who changed their beliefs from irrational to rational were able to reduce social and sports-specific anxiety [57,58]. The effect of rational beliefs therapy on athlete motivation has also been proven [59]. Professional athletes who are in a results-focused environment are under constant pressure to perform at their best, and also under the pressure of potentially losing funding. Therefore, for an athlete, the belief from a rational ‘want to’, expressing desire, can easily change to an irrational ‘have to’, expressing need [60]. Consequently, the athlete’s anxiety level will also increase [53].

In accordance with the literature and the data presented [5,6,53,57], an athlete’s performance may also be affected by their perception of the supporters. It seems that the constant pressure that fans exert on athletes may contribute to the activation of irrational beliefs and, thus, increase anxiety and stress levels. On the other hand, it is possible that support and understanding from the spectators will lead to the dominant formation of rational beliefs. To our knowledge, previous studies have not analyzed how athletes perceive the behavior of supporters, especially in the context of rational and irrational beliefs and the emotions experienced by athletes during sports competitions. Considering the evolutionary tendency to focus attention on negative, threatening triggers [61], players with dominant negative beliefs about spectators will be more sensitive to their behavior. In the present study, we examined the relationship between athletes’ sensitivity to the positive and negative behavior of supporters and a variety of psychological factors. The aim of the study is to see if there is a relationship between athletes’ sensitivity to the positive and negative behavior of supporters and traits such as anxiety and stress or the strategies used to cope with stress.

It was hypothesized that:(1)Athletes with higher levels of sports anxiety are more sensitive to the behavior of the supporters.(2)Athletes with higher levels of susceptibility to stress are more sensitive to the behavior of the supporters.(3)The athletes’ levels of sports anxiety, fear of negative evaluation, sensitivity to positive and negative behavior of the supporters, stress susceptibility, and coping strategies are related to their age and sex. The younger the athletes, the higher their level of stress susceptibility, sports anxiety, and fear of negative evaluation and the weaker the coping strategies. Female athletes are more sensitive to the positive and negative behaviors of the supporter, and have a higher level of fear of negative evaluation than the male athletes.(4)Athletes’ sensitivity to the positive and negative behaviors of supporters differs in terms of affective predictors, i.e., their level of sports anxiety, fear of negative evaluation, and their field position.

## 2. Materials and Methods

### 2.1. Subjects

The sample consisted of 171 professional athletes, that is, athletes belonging to professional sporting associations and working full-time as an athlete [62]. In total, 86 subjects (50%) were football players, who play at different sports levels, i.e., from the 4th league to the premier league. The sample also consisted of volleyball players, who represented 21% of the subjects (n = 36), ice hockey players representing 13% of the subjects (n = 23), as well as basketball players, representing 6% (n = 10), skateboarders, representing 6% (n = 10), and track cyclists, representing 4% (n = 6). The research included female (n = 44) and male (n = 127) athletes. The sample included 3 age groups, i.e., group 1 (16–22 years old (55%)), group 2 (23–28 years old (27%)), and group 3 (over 29 years old (18%)).

The present study was approved by the Jagiellonian University Institute of Applied Psychology Research Ethics Committee (app. no. 115/2021). The study was conducted according to the guidelines of the Declaration of Helsinki [63,64,65]. The objectives of the study and its requirements were explained to the subjects, and all participants provided their consent.

### 2.2. Assessment Tools and Data Collection

The following tests were used in the study: The Perceived Stress Scale (PSS-10) [66], The Sport Anxiety Scale-2 (SAS-2) [67], Brief Fear of Negative Evaluation II (BFNE II) [68], Athletic Coping Skills Inventory-28 (ACSI-28) [69], author’s questionnaire of athletes’ sensitivity to the positive and negative behavior of supporters [70]. The research was carried out during team training camps. The results of the study were subjected to statistical analysis.

The Perceived Stress Scale (PSS-10) [66] is a popular tool for measuring psychological stress. The scale consists of 10 questions concerning the respondent’s subjective stress assessment related to their life situation over the past month. The subject responds by writing the appropriate number next to the question, from 0, “never”, to 4, “very often.” The Cronbach’s alpha reliability coefficient is 0.86. The PSS10 can be used for the study of athletes [71].

The Sport Anxiety Scale-2 (SAS-2) [67] is a 15-item questionnaire that assesses the competitive trait anxiety experienced by athletes before or during a competition. The scale includes three factors: somatic anxiety, worry, and concentration disruption, which are related to cognitive anxiety. The Cronbach’s alpha reliability coefficient for somatic anxiety was 0.89; for worry, it was 0.91; and for concentration disruption, it was 0.84 [67]. The scale has been translated into Polish.

The Brief Fear of Negative Evaluation II (BFNE II) [68] measures how much the respondent expects to be judged negatively by others and make a negative impression on others. The respondents refer to statements on a 5-point Likert scale ranging from 1,“this does not describe me at all”, to 5, “this completely describes me”. The total score a respondent can obtain is between 12 and 60, with a higher score indicating greater fear of negative evaluation [68]. The Cronbach’s alpha reliability coefficient for the Brief Fear of Negative Evaluation Scale II is 0.95 [68]. The scale has been translated into Polish.

The Athletic Coping Skills Inventory-28 measures an athlete’s psychological coping skills in seven key areas, i.e., coping with adversity, coachability, concentration, confidence and achievement motivation, goal-setting, peaking under pressure, and freedom from worry. The ACSI-28 consists of 28 statements to which the subject refers on a 4-point scale. In each of the subscales, the subject can score from 0 to 12 points; the higher the score, the more developed the subject’s particular coping strategy. Summing up the scores of all subscales shows how developed the athlete’s ways of coping in sports situations are. The Cronbach’s alpha reliability coefficient for the ACSI-28 is 0.95 [69]. The scale has been translated into Polish.

Concerning the Author’s questionnaire of athletes’ sensitivity to the positive and negative behavior of supporters [70], the tool consists of 31 questions measured on a 5-point scale, where 1 indicates that “it does not describe me at all” and 5 indicates that “it completely describes me”. It includes two subscales: sensitivity to both the positive and negative behaviors of the supporters. The internal reliability of the subscales, as measured using Cronbach’s alpha reliability coefficient, is 0.85 for sensitivity to positive behavior and 0.88 for sensitivity to negative behavior, respectively.

## 3. Results

The IBM SPSS Statistics 27 PS IMGO PRO 7.0 software package was used for the statistical analysis of the data. The statistical analysis of the variables used Student’s *t*-test, one-way ANOVA, and stepwise multiple regression. The distribution of the quantitative variables in terms of normality was analyzed by using skewness, kurtosis, and Kolmogorov–Smirnov tests. It was assumed that if skewness was between −1 and 1, kurtosis was between −2 and 2; furthermore, if the results of the Kolmogorov–Smirnov test were not statistically significant, the analyzed quantitative variables would then have a distribution that is close to normal.

The results of descriptive statistics are presented in Table 1, Table 2 and Table 3. Regarding psycho-demographic variables, this study focused on sex and age. Table 1 shows the results of the male and female participants for the two dependent variables that measure athlete sensitivity to the positive and negative behaviors of the supporters and the level of athletes’ fear of negative evaluation (BFNE II). Sex differences were found to be statistically significant only for the variable sensitivity to positive behavior of the supporters (SPS) and BFNE II. The male athletes scored higher than female athletes in SPS (t = 49.33, *p* = 0.001). A parallel relationship was also found for the BFNE II. The male athletes, in comparison to the female athletes, perceive a higher level of BFNE II (t = 26.64, *p* = 0.001). As for the sensitivity to negative behavior of the supporters (SNS), the differences for the sex variable are only at the level of trend (t = 7.91, *p* = 0.08), and males are more sensitive than females.

Regarding the age of the players, the results are shown in Table 2. Statistically significant differences were found only in the case of SPS (F(2.168) = 3.52, *p* = 0.03). The highest levels of SPS were obtained by the group 1 athletes (M = 40.78, SD = 9.51). On the other hand, the lowest level of SPS was obtained by the group 3 athletes (M = 35.7, SD = 8.35). The post hoc analysis, which was carried out using Dunnett’s T3 Test (which does not assume homogeneity of variance), showed that a significant difference applies to athletes from groups 1 and 3 (*p* = 0.01) in the studied sample. The difference between groups 1 and 2, as well as the one between groups 2 and group 3, did not reach the level of statistical significance, despite nearing it. Age did not turn out to be a differentiating variable for SNS (F(2.168) = 2.31, *p* = 0.11).

When analyzing the differences between the athletes’ age and the other explanatory variables included in our study (Table 2), it should be noted, that statistically significant differences were detected only for four variables: BFNE II (F(2.168) = 5.73, *p* = 0.003), Sport Anxiety (SAS-2): worry ((F(2.168) = 3.212, *p* = 0.04); perceived stress (PSS-10) (F(2.168) = 4.259, *p* = 0.01); and Athletic Coping Skills (ACSI-28), freedom from worry (F(2.168) = 4.933; *p* = 0.01).

As for BFNE II, the highest scores were obtained by group 1 (M = 25.98, SD = 11.61), while the lowest scores were obtained by group 3 (M = 20.4, SD = 9.03). Statistically significant differences between age groups in BFNE II were detected only between group 1 and group 2 athletes (*p* = 0.01) and group 1 and group 3 ones (*p* = 0.03).

Within the variable SAS-2, worry, the highest scores were obtained by group 1 (M = 9.51, SD = 3.82). However, post hoc analysis showed that these differences applied only to group 1 and group 2 (*p* = 0.04). However, no differences were found between group 1 and group 3, as well as between group 2 and group 3 athletes. In terms of the differences in *PSS-10*, the highest level was obtained by group 1 (M = 18.17, SD = 7.29) and the lowest by group 2. Post hoc analysis showed that the difference was statistically significant (*p* = 0.02) only between these two age groups. ACSI-28, freedom from worry, was another variable that differentiated the age of the studied athlete groups. The highest level of ASCI-28, freedom from worry, was obtained by group 2 (M = 8.26, SD = 2.43), and the lowest was obtained by group 1 (M = 6.85, SD = 2.66). However, statistically significant differences, as a result of post hoc analysis (using Dunnett’s T3 test), were detected only between group 1 and group 2 athletes (*p* = 0.09).

The differences between the three groups were also detected for ACSI-28 coping with adversity (F = (2, 168) = 1.879, *p* = 0.08) and ASCI-28 confidence and achievement motivation (F(2, 168) = 1.673, *p* = 0.08), but only at the trend level. The highest scores of the ACSI-28 coping with adversity variable were obtained by the group 3 athletes (M = 7.71; SD = 2.46), and for the ACSI-28 confidence and achievement motivation variable, these were obtained by the group 2 athletes. Since there was no statistically significant difference between the results, we did not analyze them in further detail.

It should be mentioned that no statistically significant differences were detected among the different age groups of athletes with regard to the other variables included in the study. This applies to the variables related to anxiety, i.e., SAS-2 somatic anxiety (F(2, 168) = 1.163, *p* = 0.32) and SAS-2 concentration disruption (F(2, 168) = 1.168, *p* = 0.28), as well as the ones coping with stress, i.e., ACSI-28 coachability (F(2, 168) = 1.398, *p* = 0.29), ACSI-28 concentration (F(2, 168) = 0.181, *p* = 0.89), ACSI-28 goal setting (F(2, 168) = 0.512, *p* = 0.62), and ACSI-28 peaking under pressure (F(2, 168) = 0.012, *p* = 0.98). Therefore, hypothesis 3 was partially confirmed but only for group 1 and 3 athletes.

The main aim of the study was to detect predictors of athlete sensitivity to both the positive (SPS) and negative (SNS) behaviors of the supporters. For this purpose, a multiple regression analysis was conducted using a stepwise method. Four groups of independent variables were selected for the analysis: (A) psychological stress (PSS-10), (B) fear of negative evaluation (BFNE II), (C) sports anxiety (SAS-2), and (D) coping with stress (ACSI-28). In total, there were 12 variables. The results are shown in Table 4 and Table 5.

Regarding the predictors of SPS, only three variables entered the model, i.e., ACSI-28 freedom from worry (β = 0.27, *p* = 0.001), ACSI-2*8* coachability (β = −0.28; *p* = 0.001), and ACSI-28 confidence and achievement motivation (β = −0.21, *p* = 0.001). These variables explain a total of 15% of the variance in SPS (Table 4). Two other variables, i.e., SAS-2 worry (β = 0.14, *p* = 0.06) and ACSI-28 concentration (β = 0.15; *p* = 0.06), were also on the edge of statistical significance.

As for the analysis of the SNS, on the other hand, only two explanatory variables entered the model as predictors, i.e., ACSI-28 freedom from worry (β = 0.33, *p* = 0.001) and BFNE II (β = 0.31, *p* = 0.001). The two predictors explain 31% of the variance in the SNS dependent variable (Table 5).

In terms of the predictors of athlete sensitivity to spectator affective responses among the four groups of independent variables (A, B, C, and D), depending on the athletes’ field position (Table 4), the multiple regression with the stepwise method was used independently for each of the dependent variables. The football, basketball, and volleyball players were divided by their positions on the field. The ice hockey players were not included due to the lack of a position equivalent to a midfielder in this sport. The athletes of individual sports were also not included. As for the SPS, the highest percentage of variance explained was obtained by midfielders (38%), followed by forwarders (32%), and the lowest was by defense (only 9%). Three variables proved to be predictors of SPS for midfielders, i.e., PSS-10 psychological stress (β = −0.46, *p* = 0.001), ACSI-28 peaking under pressure (β = −0.45, *p* = 0.001), and ACSI-28 concentration (β = −0.33, *p* = 0.02). For forwarders, SAS-2 worry (β = −0.52, *p* = 0.001) and ACSI-28 goal-setting (β = −0.180, *p* = 0.02) entered the model, respectively. For defenders, only one variable, i.e., ACSI-28 confidence and achievement motivation, was found to be a predictor of SPS (β = 0.32, *p* = 0.02).

The SNS variable different prediction models were obtained with four groups of independent variables (A, B, C, and D) relative to the position of the athletes (Table 5). The highest percentage of variance explained with SNA was obtained by defenders (40%). Its predictors are ACSI-28 freedom from worry (β = −0.43, *p* = 0.001), ACSI-28 coachability (β = −0.34, *p* = 0.001), and BFNE II (β = 0.23, *p* = 0.05). A slightly lower value of the explanatory variance of SNS was obtained by forwarders (26%), whose strong predictor turned out to be BFNE II (β = 0.54, *p* = 0.001). The lowest value of the variance explained in SNS was in midfielders. In this case, only ACSI-28 freedom from worry entered the prediction model (β = 0.41, *p* = 0.02). Hypothesis 4 was confirmed.

## 4. Discussion

The aim of our research was to determine the relationship between the sensitivity levels of the athletes to the behavior of spectators during sports competitions and psycho-demographic variables, as well as a variety of psychological factors, such as the level of anxiety and stress and the coping with stress skills of athletes across different disciplines. Previous studies [1,2,3,4,7,8] regarding the impact of spectators on sports performance have mostly been based on the athletes’ level of performance. However, there is a lack of studies that address the subject of the sensitivity of an athlete to the affective behavior of supporters.

In the context of psycho-demographic variables, we noted that men are more sensitive to the positive behavior of supporters than women. This means that for men, being cheered on by supporters is extremely important. Male athletes also showed higher levels of fear of negative evaluation. However, the obtained results do not coincide with previous research [72], as female athletes were observed to have a higher level of fear of negative evaluation than male athletes [73] and anxiety related to the social evaluation of their appearance [39], i.e., two variables that explain the need for approval from the spectators. Men’s high sensitivity to the positive behavior of fans should be understood as a strong need for their acceptance by the public. On the other hand, it is not excludable that men’s higher sensitivity to positive spectator behavior might also be influenced by their dispositional characteristics. In our study, this characteristic was not measured.

Our study also showed that the youngest players are more sensitive to positive fan behavior by a significantly higher degree than the oldest players. This was also true for the level of fear of negative evaluation, which was highest in the youngest athletes and lowest in the oldest athletes. Both of these results are most likely related to the athletes’ experience at the competition level. Young athletes are not as experienced with competing; therefore, they pay more attention to spectator support. In a meta-analysis, Strauss [48] proved that the level of top athletes’ performance with spectator participation will be higher than that of inexperienced athletes. However, on the other hand, it has been proven that supportive spectator behaviors lead to an increase in performance quality [7], which may result in athletes associating their better performance with positive spectator behavior. The oldest athletes are more experienced, therefore, we can also assume that they are also accustomed to competitive situations, which means they no longer pay as much attention to the cheering of spectators.

Another significant result obtained for the youngest players was a strong tendency to worry, i.e., a component of cognitive anxiety. It is worth noting here that the lowest level of worrying was achieved by middle-aged players. A similar distribution of results was also obtained for the stress variable. Additionally, the youngest athletes turned out to be the most susceptible to stress, while middle-aged athletes turned out to be the least susceptible to stress. The results indicate that the athletes who already have some experience and are still not approaching the end of their sporting careers are the least worried and stressed of all. It is also worth noting here that the age range defined in our study as average (23–28 years old) is the age at which a player reaches peak performance. According to Dendir [74], soccer players reach peak performance at the age of 25–27 years old. Similarly, hockey players reach peak performance between the ages of 27 and 29 years old [75,76], basketball players between the ages of 27–28 [77], and volleyball players between the ages of 27–29 [78].

An interesting result that was also obtained in our study, in relation to the age of the athletes, refers to the strategies used to cope with stress. Middle-aged athletes achieved the highest level of freedom from worry, while the youngest athletes achieved the lowest. It might seem that this could be related to the experience of an individual athlete but given that it is not the oldest athletes who achieve the highest levels of freedom from worry, we hypothesize that this could be related to other variables. An influential factor might be whether the athlete feels confident in the team or is at their peak performance [74]. It may also be significant that athletes between the ages of 26 and 30 years old have the highest market value [79]. Therefore, it seems that they may feel financially secure and stable. However, the confirmation of these hypotheses requires further research.

The main aim of the research was to find predictors of athlete sensitivity to both the positive or negative behaviors of supporters. Based on the results of the study, we determined three variables to be predictors of athlete sensitivity to positive supporter behavior. Athletes who are sensitive to the positive behavior of supporters are characterized by strategies for coping with stress, such as coachability, confidence, and achievement motivation, and they also lack freedom from worry. This means that an athlete who is sensitive to the positive behavior of supporters is a person who is open-minded and accepts constructive criticism and who consistently works and puts maximum effort into their performance. On the other hand, they also have a low level of freedom from worry; that is, they worry about weak performance, fearing what others will think if they make a mistake. A low level of freedom from worry is also a predictor of sensitivity to the negative behavior of supporters. The results seem to be consistent with the assumptions of the freedom from worry scale [44]. An athlete with a high score on this scale does not worry about the opinion of others concerning their performance; the most important thing for them is to be satisfied with themselves, while an athlete who worries about the opinion of fans about themselves will simultaneously be more sensitive to their positive and negative behavior. The second predictor of sensitivity to the negative behavior of supporters was found to be fear of negative evaluation. This result seems to be consistent with Cottrell’s [80] fear of evaluation theory. Individuals with a high level of fear of negative evaluation feel that they will be judged; therefore, they are more sensitive to supporter behavior, especially to one, which could indicate a negative perception of the athlete.

In terms of preparing an athlete for the competition, we look not only at their physical preparation but also at their psychological preparation. Having in mind the appropriate selection of players for the team, coaches should pay attention to the psychological characteristics that are specific to talented players [81]. Additionally, scouts include in their scouting sheets not only scales for assessing a player’s physical skills but also mental ones [82]. According to the research, team sports athletes playing in different positions have different physical [83,84,85] and psychological skills [86,87,88,89]. For this reason, we decided to look for predictors of athlete sensitivity to the behavior of supporters according to their positions on the field. The athletes were divided into three positions on the field: midfielders, defenders, and forwarders. In terms of sensitivity to the positive behavior of supporters for midfielders, level of susceptibility to stress and the two strategies of coping with stress were found to be predictors, i.e., peaking under pressure and concentration. This means that midfielders, who are sensitive to the cheering of fans, will show higher levels of susceptibility to stress and the ability to cope with pressure and maintain maximum concentration. It can be assumed that, for these athletes, the pressure created by cheering spectators and the stress associated with it will be perceived as a challenge rather than a threat. The cheering of spectators will serve as an encouragement to take up the challenge. The predictors of sensitivity to positive behavior of supporters for forwarders were found to be worry and coping with stress through goal-setting. Forwarders, who cope with stress by mentally preparing for the game and planning a good performance will be more sensitive to spectators cheering. For the defenders, a predictor of sensitivity to the positive behavior of supporters was found to be coping with stress through confidence and achievement motivation. This may be due to the need and search to be rewarded by spectators for their efforts.

Definitely, different predictors were obtained in the analysis context of sensitivity to the negative behavior of supporters. For defenders, it was a low level of freedom from worry and coachability and a high level of fear of negative evaluation. Defenders, who are sensitive to negative spectator behavior are characterized by experiencing fear of negative evaluation; they do not accept criticism and will be significantly worried about weak performance and mistakes, especially in the sense of what others will think of them. Forwarders who are sensitive to the negative behavior of the public will, at the same time, feel fear of negative evaluation. The midfielders, on the other hand, are characterized by worrying about weak performance.

The research carried out can be applied, as it shows how different psychological variables are predictors of athletes’ sensitivity to spectator behavior, depending not only on the athletes’ individual characteristics [46,67] and their susceptibility to stress [44,66] but also the position they play during competitions. Therefore, mental preparation must be individualized with regard to the athlete’s position on the field. The findings we obtained in this analysis are consistent with the research on two motivation strategies—striving for success and avoiding failures [90]. It is also supported by Grey’s theory of individual dispositions [91].

### Limitations and Future Directions

Certain limitations of this study should be kept in mind for further research. First of all, a significant limitation of our study seems to be the small number of athletes participating in individual sports who were examined. The study focused mainly on team sports athletes, and they made up the great majority of the subjects. By studying a similar number of athletes involved in individual and team sports, we could compare these two groups. This seems interesting given the difference in the age at which the individual athletes reach peak performance compared to the team athletes. In individual sports such as swimming, the age for reaching peak performance is 20 to 27 [92], while in golf, it is 34 [93]. Whether an individual discipline athlete will show lower levels of worry and stress at different ages than a team discipline athlete requires further research.

## 5. Conclusions

The research found a relationship between an athlete’s sensitivity to the behavior of supporters, psycho-demographic variables, and different psychological factors. We noted that males are more sensitive to positive spectator behavior than females. The youngest athletes, compared to the oldest, are significantly more sensitive to positive behavior of the spectators and show a higher level of fear of negative evaluation. Middle-aged athletes obtain the lowest levels of worry, a component of cognitive anxiety, and are also the least susceptible to stress and cope with stress well. The study determined three variables that are predictors of athlete sensitivity to the positive behavior of supporters. The athletes, who are sensitive to the positive behavior of supporters are characterized by strategies for coping with stress, such as high levels of coachability, confidence and achievement motivation, and low levels of freedom from worry. Predictors of sensitivity to the negative behavior of supporters are a low level of freedom from worry and a high level of fear of negative evaluation. In the context of the athlete’s position, the predictors of sensitivity to the positive behavior of supporters are for the midfielders: the high level of susceptibility to stress and two strategies of coping with stress, i.e., peaking under pressure and concentration. Forwarders, who are sensitive to the behavior of supporters, are characterized by high levels of worry and coping with stress through goal-setting, while for defenders, the way of coping with stress is confidence and achievement motivation. In terms of sensitivity to the negative behavior of supporters, for defenders, the predictors are low levels of freedom from worry, coachability, and a high level of fear of negative evaluation. Forwarders, who are sensitive to the negative behavior of supporters, are characterized by fear of negative evaluation.

The coordination of communication between coaches, medical and psychological staff at all stages of training and competition is particularly important in the athlete preparation process [94,95]. The individualization of psychological intervention is significant as well [96]. The practical effect of the research is the possibility to identify the directions of personalized treatment of athletes in the process of training and their psychological preparation for competitions. The results of the research can be part of the training process to be developed in the following years.

## Figures and Tables

**Table 1 ijerph-20-06084-t001:** Basic descriptive statistics of athletes related to athletes’ sex for variables BFNE II, SNS, and SPS.

Scale	Feale			Male			Whole Sample		
	N	Mean	SD	N	Mean	SD	N	t	*p*
BFNE II	44	21.0	6.5	127	30.2	9.8	171	26.64	0.001
SNS	44	39.4	10.4	127	41.6	11.5	171	7.91	0.08
SPS	44	37.8	9.6	127	42.8	8.9	171	49.33	0.001

Note: BFNE II, fear of negative evaluation; SPS, sensitivity to positive behavior of the supporters; SNS, sensitivity to negative behavior of the supporters.

**Table 2 ijerph-20-06084-t002:** Means and standard deviations as well as the analysis of variance of athletes’ age and position on the SAS-2 and PSS-10 scales.

Scale		SAS-2 Worry		SAS-2 Somatic Anxiety		SAS-2 Concentration Disruption		PSS-10	
	N	Mean	SD	Mean	SD	Mean	SD	Mean	SD
Defender	57	8.86	0.84	7.11	1.82	7.47	2.75	17.36	7.31
Forward	33	9.00	0.63	7.15	3.06	7.57	2.50	15.36	7.63
Midfielder	42	8.42	0.56	6.91	1.91	7.43	2.21	17.26	6.93
Whole sample	132	8.76	0.67	7.11	2.21	7.48	2.51	16.83	7.26
F		0.264		0.892		0.114		1.144	
Df		2/129		2/129		2/129		2/129	
*p*		n.s.		n.s.		n.s.		0.08	
Group 1	95	9.51	3.82	7.33	2.52	7.71	2.73	18.17	7.29
Group 2	46	7.95	2.59	6.76	1.68	7.08	1.77	14.57	5.91
Group 3	30	8.83	3.11	7.40	1.91	7.26	1.92	16.66	6.98
Whole sample	171	8.97	3.46	7.19	2.23	7.46	2.38	16.93	7.02
F		3.212		1.163		1.168		4.259	
Df		2/168		2/168		2/168		2/168	
*p*		0.04		n.s.		n.s.		0.01	

Note: SAS-2, sport anxiety; PSS-10, psychological stress; group 1, 16–22 years old; group 2, 23–28 years old; group 3, over 29 years old.

**Table 3 ijerph-20-06084-t003:** Means and standard deviations as well as the analysis of variance of athletes’ age and position on theACSI-28 scale.

Scale		ACSI-28 Coping with Adversity		ACSI-28 Coachability		ACSI Concentration		ACSI-28 Confidence and Achievement Motivation		ACSI-28 Goal-Setting		ACSI-28 Peaking under Pressure		ASCI-28 Freedom from Worry	
	N	Mean	SD	Mean	SD	Mean	SD	Mean	SD	Mean	SD	Mean	SD	Mean	SD
Defender	57	7.17	2.11	8.98	2.11	7.65	2.24	8.29	2.17	6.42	2.28	7.47	2.57	7.28	2.78
Forward	33	7.48	2.82	9.27	2.58	8.27	2.33	8.72	2.26	7.09	2.69	7.66	3.07	7.66	2.61
Midfielder	42	7.16	2.64	9.07	2.57	7.85	2.48	8.52	2.17	6.71	2.48	7.35	2.94	7.88	2.41
Whole sample	132	7.25	2.45	9.08	2.35	7.87	2.34	8.47	2.19	6.68	2.45	7.48	2.81	7.56	2.62
F		0.198		0.157		0.738		0.411		0.783		0.111		0.661	
Df		2/129		2/129		2/129		2/129		2/129		2/129		2/129	
*p*		n.s.		n.s.		n.s.		n.s.		n.s.		n.s.		n.s.	
Group 1	95	6.91	2.22	8.71	2.60	7.67	1.95	8.47	2.03	6.94	2.35	7.24	2.76	6.85	2.66
Group 2	46	7.54	2.71	9.41	2.07	7.89	2.81	8.71	2.34	6.69	2.35	7.21	2.62	8.26	2.43
Group 3	30	7.71	2.46	8.86	1.69	7.60	2.66	7.83	2.26	6.46	2.58	7.16	2.58	7.66	2.36
Whole sample	171	7.22	2.42	8.92	2.33	7.71	2.33	8.42	2.16	6.79	2.39	7.22	2.68	3.37	2.61
F		1.879		1.398		0.181		1.673		0.512		0.012		4.933	
Df		2/168		2/168		2/168		2/168		2/168		2/168		2/168	
*p*		0.08		n.s.		n.s.		0.08		n.s.		n.s.		0.01	

Note: ACSI-28, athletic coping skills; group 1, 16–22 years old; group 2, 23–28 years old; group 3, over 29 years old.

**Table 4 ijerph-20-06084-t004:** Regression coefficients for explanation of athletes’ sensitivity to the positive behavior of the supporters related to athlete position.

Scale	SPS		Defenders		Forwarders		Midfielders	
	t	β	t	β	t	β	t	β
SAS-2 worry	1.69	0.14 ^	0.74	0.09	3.46	0.52 ***	1.10	0.15
ACSI-28 coachability	3.29	0.27 ***	0.88	0.11	0.76	0.12	0.09	0.01
ACSI-28 concentration	1.79	0.15 ^	1.09	0.17	−0.23	−0.04	2.26	0.33 **
ACSI-28 confidence and achievement motivation	2.73	0.21 ***	2.53	0.32 **	1.06	0.19	0.39	0.06
ACSI-28 goal-setting	0.54	0.04	0.27	0.03	2.21	0.33 **	0.58	0.08
ACSI-28 peaking under pressure	1.34	0.12	0.44	0.06	0.65	0.14	3.12	0.45 ***
ASCI-28 freedom from worry	−3.56	−0.28 ***	−0.02	−0.00	−0.48	−0.09	−0.99	−0.13
PSS-10	0.51	0.04	−0.71	−0.09	0.28	0.05	3.41	0.46 ***
	Adj R^2^ = 0.17; Change R^2^ = 0.15		Adj R^2^ = 0.11; Change R^2^ = 0.09		Adj R^2^ = 0.36, Change R^2^ = 0.32		Adj R^2^ = 0.42, Change R^2^ = 0.38	
	F of change = 9.78, *p* of change < 0.001		F of change = 6.39, *p* of change = 0.01		F of change = 7.99, *p* of change = 0.001		F of change = 9.23, *p* of change < 0.001	

Note: SAS-2, sports anxiety; PSS-10, psychological stress; ACSI-28, athletic coping skills; SPS, sensitivity to positive behavior of the supporters. *p* = 0.001 ***, *p* = 0.02 **, *p* = 0.06 ^.

**Table 5 ijerph-20-06084-t005:** Regression coefficients for explanation of athlete sensitivity to the negative behavior of the supporters related to athlete position.

Scale	SNS		Defenders		Forwarders		Midfielders	
	t	β	t	β	t	β	t	β
BFNE II	3.76	0.31 ***	1.97	0.23 *	3.43	0.54 ***	1.03	0.20
ACSI-28 coachability	−1.90	−0.13	−3.05	−0.34 ***	−1.13	−0.17	−0.49	−0.08
ASCI-28 freedom from worry	−4.11	−0.33 ***	−3.86	−0.43 ***	−1.18	−0.29	−2.81	−0.41 ***
	Adj R^2^ = 0.32; Change R^2^ = 0.31		Adj R^2^ = 0.42; Change R^2^ = 0.40		Adj R^2^ = 0.29; Change R^2^ = 0.26		Adj R^2^ = 0.17; Change R^2^ = 0.15	
	F of change = 38.56, *p* of change <0.001		F of change = 19.34, *p* of change < 0.001		F of change = 11.78, *p* of change = 0.001		F of change = 7.85, *p* of change = 0.007	

Note: BFNE II, fear of negative evaluation; ACSI-28, athletic coping skills; SNS, sensitivity to negative behavior of the supporters. *p* = 0.001 ***, *p* = 0.05 *.

## Data Availability

Not applicable.

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
