# Peer review of "Sport Anxiety, Fear of Negative Evaluation, Stress and Coping as Predictors of Athlete’s Sensitivity to the Behavior of Supporters"

_ijerph, 2023, doi:10.3390/ijerph20126084_

Round 1
Reviewer 1 Report
Dear Authors,
First of all, I would like to congratulate you for all the work you have done in this research. This is an interesting topic, with clinical impact and of which, undoubtedly, more research is needed to provide valuable knowledge to the area.
However, in the manuscript there are shortcomings and formal errors that should be resolved before its possible publication in this Journal.
Abstract:
The transmission of results could be done in a more objective and reliable way (less qualitative).
The Conclusions subsection should be complemented with an interpretation of its clinical impact.
Introduction:
There is an abuse of the use of bibliographic references. That is to say, the authors should choose the most representative, recent ones and those that contribute the greatest value of new knowledge to the subject.
However, I miss the use of other recent references of great value such as, for example: 10.3390/healthcare10122357
In addition, the Authors should try to provide a more succinct version of this section.
Material and Methods:
Section 2.1. does not refer in its content to the title of the section. Rewrite, please.
This entire section, in general, is out of order and has typos and grammatical errors. Please rewrite and correct thoroughly.
Authors should pay special attention to transmit all the information in an orderly manner and, for this purpose, it is of great value and useful that you use the appropriate EQUATOR Network checklist.
Results:
The Results provided can also be rewritten to make them easier and clearer to read.
The statistical techniques applied should be complemented with the use of effect sizes to provide real information on their value.
On the other hand, the values of statistical representativeness and statistical power of the sample size used should be transmitted.
The descriptive statistical values should be transmitted with only one decimal place (with none if it is a zero).
The Authors, again, should try to provide a less verbose and extensive version of the text of this section and look for alternative ways (more graphic and visual) to obtain information about the data obtained.
Discussion and Conclusions:
The Authors should limit their generalizations and even small lucubrations that they make in Discussion. But, in general, they are correct sections.
Kind regards
Reviewer 2 Report
This manuscript describes an observational study that utilized questionnaires to assess several psychological characteristics of athletes to understand predictors of how athletes perceive the behavior of spectators. This is an interesting study that adds to the field, though there are some limitations and areas that could be improved by clarification.
Major concerns:
The authors need to describe the specific statistical analyses that were used for each comparison. When did the authors use a Student’s t test or a one-way ANOVA? There were so many comparisons made that the authors should use an FDA correction, especially for t-tests.
The first paragraph of the results presents the findings from comparing sexes, though the authors never mention sex differences in the introduction or the methods. Sex is not mentioned in the four hypotheses. If sex differences are included, then it would be appropriate to use a 2-way ANOVA with sex as a factor and age group as another factor.
The authors sometimes include statistics (e.g. F values) in the results, yet at other times they do not.
Minor concerns:
In the introduction, the authors state that “competitors are professionals”. I recognize that the competitors in this sample are professionals, but not all competitors are professionals, so the authors should be clear that their sample is composed of professionals and that the results may not extend to the majority of competitors.
A table with subject demographics would be helpful to understand the population better.
Table 2. could be split into two separate tables since it is presenting two different comparisons in the same subjects.
The position specific analysis is limited to only the football players. The authors should justify this analysis and the exclusion of the other athletes.
The discussion of age of peak performance in the discussion section is very specific to football and hockey players, which only represented 63% of their population. This is briefly mentioned in the limitations section, but if the authors are going to use age of peak performance as an explanatory factor, they should mention the other sports as well, especially volleyball and basketball.
The manuscript needs to be proofread and edited as there are a few typos and errors in punctuation.
Reviewer 3 Report
The study examines the
Sport Anxiety, Fear of Negative Evaluation, Stress and Coping 2 as Predictors of Athlete’s Sensitivity to the Behavior of Supporters
Comments:
· The abstract should be better written, i.e. the introduction should be state that objective was to examine whether anxiety, fear, stress, etc.. are associated with Athlete’s Sensitivity to the Behavior of Supporters
· The abstract should state the study design
· The whole abstract is hard to comprehend
· In the introduction, the hypothesis are written in a very non-understandable fashion to me
· The methods section: the authors reported design in the place of design, even the data analysis is poorly written
· The tables should have informative legends detailing the analysis done, etc…
· The tables should be professional
· While going through the results, some unrelated sentences are found: “The main aim of the study, was to detect predictors of the athletes' sensitivity to both 279 positive, (SPS) and negative (SNS) behavior of the supporters”
· Based on what: “Four groups of inde-281 pendent variables were selected for the analysis: (A) the psychological stress (PSS), (B) 282 fear of negative evaluation (BFNE), (C) sport anxiety (SAS), and (D) coping with stress 283 (ACSI). In total, 13 variables.. what???”
· Table 3. The predictors of athletes' SPS, due to athlete position
· In regards to table 3, how can we determine predictors without Odds ratio???
· The introduction and discussion can be more concise and informative
The entire manuscript need a comprehensive professional writing and editing
Reviewer 4 Report
The research topic is undoubtedly relevant, but reading the text raises a number of questions.
The novelty of the work looks rather doubtful, given that the study is devoted to a topic that has been discussed many times and even to a simple layman it seems intuitively obvious (the behavior of fans affects an athlete - this is practically an axiom)
The interpretation of the results raises questions. In particular, in the discussion of gender differences: (Line 331).
In the context of psycho-demographic variables, we noted that men are more sensitive to the positive behavior of supporters, than women. This means, that for men, being cheered on by supporters is extremely important. However, the obtained results do not coincide with previous knowledge [71], as it were female athletes who were observed to have a higher level of fear of negative evaluation than male athletes [72] and anxiety related to social evaluation of appearance [39], i.e. two variables that explain the need for approval from the spectators .
First of all, where is the contradiction here? Secondly, it is not clear why, in an attempt to explain the dependence on praise or disapproval, to dig so deeply, if we are talking about basic human strategies for achieving results, where some are focused on striving for success, while others are focused on avoiding failures. Undoubtedly, the recommendations given below take place, and when preparing athletes, psychological characteristics must be taken into account - but everything starts with a much more "primitive" level, which is not explained by the position of the player in the team alone (striker or defender).
In general, the conclusions look very primitive - superficial where it would be necessary to dig deeper, and unreasonably "piled up" where it is inappropriate
Round 2
Reviewer 1 Report
Dear Authors,
You have done a great job of improving and correcting the manuscript. As a result of these changes, the scientific and formal quality of the text has improved considerably.
Now I consider that the manuscript is suitable for publication in this Journal.
Kind regards
Reviewer 3 Report
Thank you for taking the comments into consideration
Reviewer 4 Report
The authors have significantly revised the text of the article. In particular, they really removed the contradiction in the analysis of the behavior of men and women, and thus this claim of the reviewer is removed.
At the same time, from the new version of the text, it is still not particularly clear what exactly the novelty of the study is.
In the text, they then say that "The main aim of the research was to find predictors of athlete sensitivity to positive or negative behavior of supporters." - but why then in the abstract "Abstract: The aim of the study was to determine a relationship between athletes' sensitivity to the 10 positive, and negative behavior of supporters, and traits such as anxiety, stress, or strategies to cope with stress." - let them immediately say that the main goal is to identify predictors, and then addiction - it's already for everyone obvious.
In terms of justifications: they talk well about strategies for overcoming stress, but for some reason they do not mention banal motivation at all, where there are also 2 main strategies - striving for success and avoiding failures. In the context of their research, this would also be appropriate, given that the motivation of an athlete (in preparation) is primary.
